# Advanced Nanotechnology-Based Nucleic Acid Medicines

**DOI:** 10.3390/pharmaceutics16111367

**Published:** 2024-10-25

**Authors:** Noriko Miyamoto, Mina Sakuragi, Yukio Kitade

**Affiliations:** 1Department of Applied Chemistry, Faculty of Engineering, Aichi Institute of Technology, 1247 Yachigusa, Yakusa-cho, Toyota 470-0392, Japan; 2Department of Materials Chemistry, Graduate School of Engineering, Aichi Institute of Technology, 1247 Yachigusa, Yakusacho, Toyota 470-0392, Japan; 3Department of Nanoscience, Faculty of Engineering, Sojo University, 4-22-1, Ikeda, Nishi, Kumamoto 860-0082, Japan; 4e-NA Biotec Inc., 3-1-2 Inabadori, Gifu 500-8043, Japan

**Keywords:** nucleic acid medicine, siRNA, miRNA, mRNA, DNA, ASO, CpG-ODN, pDNA, drug delivery system, nanotechnology, lipid nanoparticles, nanotechnology-based medicine

## Abstract

Nucleic acid medicines are a highly attractive modality that act in a sequence-specific manner on target molecules. To date, 21 such products have been approved by the Food and Drug Administration. However, the development of nucleic acid medicines continues to face various challenges, including tissue and cell targeting as well as intracellular delivery. Numerous research groups are addressing these issues by advancing the development of nucleic acid medicines through nanotechnology. In countries other than Japan (including Europe and the USA), >40 nanotechnology-based nucleic acid medicines have been tested in clinical trials, and 15 clinical trials are ongoing. In Japan, three phase I trials are ongoing, and future results are awaited. The review summarizes the latest research in the nanotechnology of nucleic acid medicines and statuses of clinical trials in Japan, with expectations of further evolutions.

## 1. Introduction

In the past two decades, nucleic acid medicines have emerged as a novel pharmaceutical modality, following small-molecule and antibody drugs. The Food and Drug Administration (FDA) has approved 21 nucleic acid medicines (Figure 1a), with future market growth expected. Since the approval of the first nucleic acid medicine in 1998, development in this field has encountered considerable challenges.

Nucleic acid medicines include antisense oligo nucleic acids (ASOs); small interfering RNA (siRNA) and microRNA (miRNA), both of which are involved in gene silencing; mRNA; plasmid DNA (pDNA), used in supplementation gene therapy; and an unmethylated cytosine guanine motif oligonucleotide (CpG-ODN), which is an immune activator for adjuvants.

Nucleic acids are rapidly degraded by nuclease enzymes including DNase and RNase in the blood and cytoplasm, and they exhibit poor cell membrane permeability as they are high-molecular-weight molecules behaving as negatively charged polymers. Despite the advancements in chemically modified nucleic acids to evade enzymatic degradation, enhancing cell permeability and targeting specific tissues and cells remain challenging [1,2,3]. For chemical modifications, modifications of the phosphate backbone, sugar, and base moieties have been developed and introduced as phosphorothioate (PS), 2′-Fluoro(F), 2′-O-methylation RNA (OMe), 2′-O-methoxyethyl RNA (MOE), locked nucleic acid or bridged nucleic acid (LNA/BNA), and morpholino oligomer (PMO) into approved nucleic acid medicines (Figure 1b).

Approved nucleic acid medicines can be used in the chemical modification of positions in nucleic acid sequences (Figure 1c). For the RNA-induced silencing complex (RISC), which requires an antisense strand (AS) of siRNA and an argonaute protein to recognize and cleave target mRNA, siRNA should be designed with 2′-F and 2′-OMe, and methods such as standard template chemistry (STC) and enhanced stabilization chemistry (ESC) modification methods have been proposed [1,4,5,6]. 2′-F and 2′-OMe are both composed of a 2′-modification of sugar moiety. The introduction of 2′-F and 2′-OMe improves nuclease resistance and target RNA binding.

ASOs are designed as gapmer and exon-skipping types. The structure of the gapmer type consists of 2′-MOE introduced at both ends of the sequence and natural DNA in the center (Figure 1c). The 2′-MOE portion is introduced for nuclease resistance, and the natural DNA portion is involved in the recognition and cleavage of the target RNA. The other ASO in Figure 1c, the exon-skipping type, is chemically modified with PMO. PMO comprises a six-membered morpholine sugar moiety and a phosphorodiamidate phosphate moiety; moreover, it is electrically neutral. PMO binds to the target RNA but is not cleaved by RNase H, thereby regulating gene expression without cleavage.

Most of the approved drugs are administered locally, reducing the loss of free diffusion due to systemic circulation. Six approved drugs use ligand-conjugated nucleic acids to enable liver targeting. However, targets other than the liver target are still difficult to target, and improving cell permeability and targeting tissues and cells are still issues to be addressed. N-acetylgalactosamine (GalNAc)-conjugated nucleic acids at the end of 3′ sense strands are utilized in six approved medicines, leveraging asialoglycoprotein receptors expressed on cells to enhance cellular uptake [1,7,8]. However, targeting beyond the liver remains challenging.

To address these issues and advance nucleic acid medicines, there is ongoing research into various nanotechnologies, including lipid nanoparticles (LNPs) [9], synthetic polymer materials, biomaterials, and nucleic acid-based materials. Nanotechnology-based medicines are expected to improve drug efficacy through more deliberate control than the free diffusion of nucleic acid alone. The present article provides an overview of the research on nucleic acid drugs utilizing nanotechnology.

## 2. Front Runner of Nucleic Acid Delivery: LNP

LNPs are the most advanced nanotechnology-based medicine used in approved mRNA and siRNA medicines, providing a variety of knowledge regarding the material design of nanotechnology [10,11]. LNPs are created by mixing two solutions of a nucleic acid dissolved in an acetic acid buffer solution and four types of lipids (phospholipids, polyethylene glycol (PEG) lipids, ionizable lipids, cholesterol) dissolved in an organic solvent using a microfluidic device [9]. LNPs form a core of nucleic acids concentrated with ionizable lipids inside the nanoparticle, and their structure is characterized by PEG lipids exposed at the interface (Figure 2a) [12].

LNP releases nucleic acids into the cytoplasm as the pH decreases from early (pH 6.0–6.2) to late (pH 5.0–5.5) endosomes during endocytosis (Figure 2b). This release occurs when the ionizable lipids ionize as the pH decreases within endosomes [13,14]. This ionization increases ion permeation within the endosomes, destabilizing the endosomal membrane and facilitating translocation to the cytoplasm. Alternatively, protonated ionizable lipids interact with the negatively charged endosomal membrane, further destabilizing it and promoting cytoplasm translocation.

The approved nucleic acid medicines that use LNP utilize lipids containing tertiary amines discovered by Peter Cullis [9,10,15,16]. Onpattro^®^ [17,18], a treatment for transthyretin-type hereditary amyloid polyneuropathy, contains 4-(dimethylamino)butanoic acid (10Z,13Z)-1-(9Z,12Z)-octadecadien-1-yl-10,13-nonadecadien-1-yl ester (DLin-MC3-DMA) (Figure 2c and Table 1). Comirnaty^®^ and Spikevax^®^ are mRNA vaccines against coronavirus disease 2019 containing 6-((2-hexyldecanoyl)oxy)-*N*-(6-((2-hexyldecanoyl)oxy)hexyl)-N-(4-hydroxybutyl)hexan-1-aminium (ALC-0315) and 1-octylnonyl 8-[(2-hydroxyethyl)[6-oxo-6-(undecyloxy)hexyl]amino]octanoate (SM102), respectively (Figure 2c and Table 1). Chemically modified siRNA is used in Onpattro (Figure 1c). Furthermore, mRNA chemically modified with *N*^1^-Methylpseudouridine (Figure 2d) [19,20], a uridine derivative that provokes a lower immunogenic response and suppresses inflammatory responses, and 5′-Cap for translation initiation is used in Comirnaty^®^ and Spikevax^®^.

As seen in the structures of these two lipids, the bulkiness of the alkyl moiety also has a role in the pH response of LNPs. Therefore, the optimization of the ionizable lipid molecular design, considering branched alkyl chains, has been studied [21,22]. Fourteen LNP medicines, including optimized ionizable lipids with mRNAs, gene-editing RNAs, pDNAs, and ASOs, are currently in the clinical trial stage (Appendix A) [23,24]. The gene-editing RNAs use CRISPR/Cas9 technology to guide RNA and Cas9 mRNA [25,26]. Regarding the importance of LNP morphology, nucleic acid-based bleb structure releases its contents similarly to a popping bubble, increasing mRNA transfection efficiency and improving gene expression [27,28]. This brevity structure is subject to the action of the buffer solution used to condition the LNP. Ninety percent of the LNP dose accumulate in the liver, because apolipoprotein E (ApoE), a lipid transport protein present in the blood, is adsorbed into LNPs and facilitates their uptake into the liver via the low-density lipoprotein receptor [17,29,30]. This kinetic control of LNPs by protein decoration via protein adsorption in the blood has also been investigated [30]. Interactions between materials and biofluid molecules in the body are also important for the kinetics of LNPs, and the interactions and kinetics of nucleic acid molecules with proteins have been reported [31,32,33]. These interactions are weaker than the antigen–antibody interaction, but they affect the pharmacokinetics.

Selective organ-targeting (SORT) LNPs were found to improve nonliver distribution issues via optimizing the lipid compositions and controlling the surface charges of LNPs (Table 2) [34,35,36]. LNPs with a neutral surface formed by the ionizable lipid 1,2-dioleoyl-3-dimethylammonium propane (DODAP) accumulate in the liver, those with a negatively charged surface formed by the anionic phospholipid 1,2-dioleoyl-sn-glycero-3-phosphate (18PA) accumulate in the spleen, and those with a positively charged surface formed by cationic lipid 1,2-dioleoyl-3-trimethylammonium propane (DOTAP) accumulate in the lungs [35]. Lung-targeting SORT LNPs, which add a permanently cationic quaternary ammonium lipid to the fifth component (the lipid composition includes 40–60% total lipids) are known to form different protein coronas of plasma proteins from liver-targeting LNPs (with reduced enrichment of ApoE) and immunoglobulins as well, with increased enrichment of vitronectin among other proteins. OF-Deg-Lin is an ionizable cationic lipid that accumulates in the spleen [37]. The lipoplexes of the neutral cytidinyl lipid 2-(4-amino-2-oxopyrimidin-1-yl)-N-(2,3-dioleoyl-oxypropyl) acetamide (DNCA) and the cationic lipid dioleoyl-3,3′-disulfanediylbis-[2-(2,6-diaminohexanamido)] propanoate (CLD) accumulate in the spleen via intravenous (i.v.) administration (Table 2) [38]. Different routes of nanoparticle administration affect protein corona formation and alter organ accumulation [39]. After i.v. or intraperitoneal (i.p.) administration, LNPs containing cationic lipids accumulate in the liver and pancreas, respectively (Table 2) [39]. These organ-specific targeting techniques are expected to greatly advance the field of nucleic acid medicine. Although research on LNPs has progressed to clinical findings and has shown no liver accumulation in vivo, the nucleic acid loading ratio in a nanoparticle is low at <10%, whereas the “PEG dilemma” antibodies cause faster phagocytosis, which is called the accelerated blood clearance (ABC) phenomenon and which remains a future challenge [40,41].

## 3. Synthetic Polymer Materials

Polyplexes are nanoparticles formed by the electrostatic interactions between cationic synthetic polymers and nucleic acids. CALAA-01 of a polyplex is formed by a synthetic cationic cyclodextrin-based polymer (CDP) and siRNA (Figure 3a) [42]. The nanoparticle has a diameter of 75 nm and binds to the transferrin receptor of the target. CALAA-01 had undergone its first clinical trials for solid tumors in NCT00689065 (Appendix A) [43,44]. The study results have shown that it was well tolerated for initial doses of 3–30 mg/m^2^ in 15 patients. However, two patients experienced dose-limiting toxicities after 2 years.

Other clinical trials are summarized in Appendix A. Local Drug EluteR (LODER^TM^) is a biodegradable polymer poly(lactic-co-glycolic) acid (PLGA) with KRAS(G12D) siRNA for pancreatic cancer treatment in NCT1676259 [45]. MK-4621 is a cationic polymer polyethyleneimine (PEI, JetPEI^TM^) with CpG-ODN for treating tumors in NCT03739138 [46]. STP705 is a histidine–lysine copolymer (HKP) with two types of siRNAs of PTGS2 and TGF-β for treating tumors in NCT05196373. STP707 is an HKP with two types of siRNAs, Cox-2 and TGF-β, for treating tumors in NCT05037149 [47]. Unit PIC (uPIC) is formed by a Y-shaped polymer, a branched-block copolymer comprising PEG, polylysine, and siRNA. The diameter of uPIC is 18 nm. The uPIC improved blood retention and tumor accumulation and was effective for treating of subjects in pancreatic cancer and brain tumor models (Figure 3b) [48,49]. Regarding the accumulation of nanoparticles in tumor sites, particles with a size between 10 and 150 nm are known to exhibit enhanced permeability and retention (EPR) effects, allowing them to leak into and accumulate in tumor tissues, although its administration rate is minimal (<1%) [50,51,52]. The pioneering nanomedicines have contributed greatly to knowledge regarding the usefulness of nanomedicines in human clinical practice. In studies using the clinical drug Doxil^®^, which incorporates Doxorubicin (Dox) of low-molecular weight drugs into liposomes composed of lipids, nanoparticles with liver accumulation characteristic improve tumor accumulation to 12% when administered at the number of doses exceeding 10^12^ particles [53]. This threshold of the number of particles is an important parameter in the development of nanomedicines for targeting organs other than the liver. Synthetic polymers are expected to help reduce the cost of pharmaceuticals because they can be synthesized in large quantities, but the process must comply with the Chemistry, Manufacturing, and Controls (CMC) guidance that carefully regulates the distribution of polymers.

**Figure 3 pharmaceutics-16-01367-f003:**
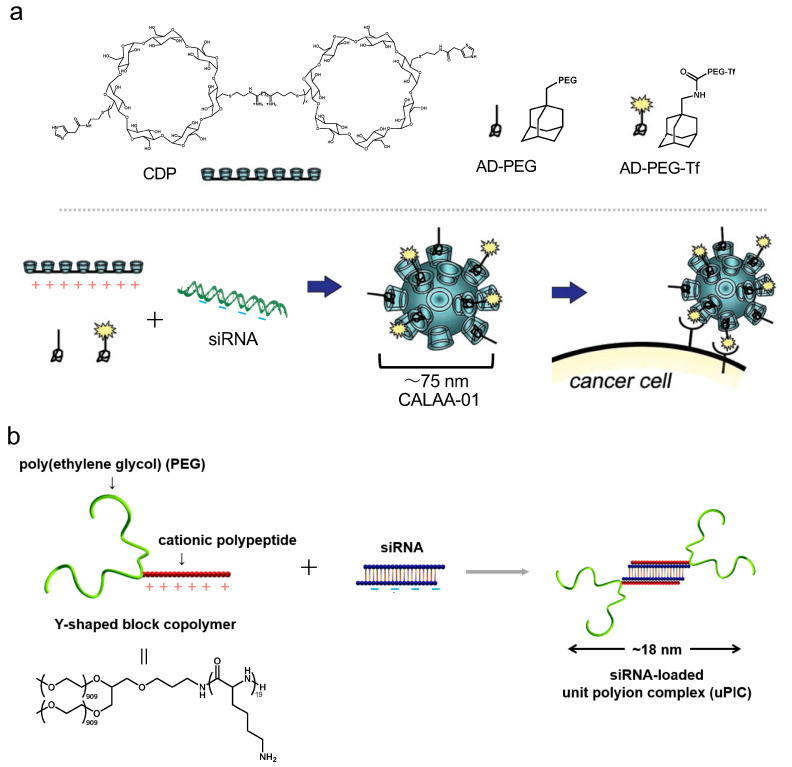
(**a**) Schematic of the CALAA-01 by edited from ref. [42] (Permission obtained from American Chemical Society) CDP, cationic cyclodextrin-based polymer; AD-PEG, adamantane-PEG; AD-PEG-Tf, AD-PEG-Transferrin (**b**) Schematic diagram of the uPIC by edited from ref. [48] (Permission obtained from Elsevier).

## 4. Biomaterials

Biomaterials, including peptides, proteins, polysaccharides, exosomes of extracellular vesicles, and nucleic acid, are used as nucleic acid carriers (the nucleic acid materials are described in Section 5). The short peptide A_6_K (AAAAAAK), which self-assembles to form a positively charged nanotube structure (Figure 4a), is complexed with nucleic acids and the pH response of the A_6_K/PTEN-siRNA complex targeting the drug resistance gene PTEN [54,55,56]. Self-adjusting α-helical coiled coils containing peptides assemble with DNA and RNA to improve the thermodynamic stability [57]. Atelocollagen, a positively charged protein, complexes with nucleic acids, improving the stability in physiological conditions. The nucleic acid delivery method using the atelocollagen/nucleic acid complex has been reported, specifically from its local administration to its systemic administration in various diseases [58,59]. Thiol-modified glycol chitosan, a polysaccharide, and thiol-modified poly siRNA, which has a high molecular weight through the thiol group, form nanoparticles [60] (Figure 4b). The combination therapy of chitosan/RAS-siRNA nanoparticles targeting the cancer-promoting gene RAS with a PTEN inhibitor induced antitumor effects [61]. The cholesterol-modified pullulan (a polysaccharide) self-assembles in water via hydrophobic interactions to form nanogels [62,63]. Nanogels can encapsulate a variety of molecules ranging from small to large nucleic acids and proteins. siRNA-encapsulated nanogel was demonstrated to downregulate target mRNA in cancer cell lines [64]. The polysaccharide schizophyllan (SPG) forms a triple-helical nanostructure through hydrogen bonding and hydrophobic interactions with polydeoxyadenosine (polydA) [65,66] (Figure 4c). CD40-siRNA-conjugated polydA and SPG complexes enable active targeting to immune cells via the dectin-1 receptor of SPG and induce RNA to interfere against CD40 [67]. The exosome functions as a communication substance between cells in the biological system [68,69]. Exosomes are natural nanoparticles and perform the active targeting delivery of nucleic acid via receptor proteins [70] (Figure 4d). To create nucleic acid medicines using exosomes, the following considerations should be considered: (1) genetic engineering methods in which cells are transfected with a plasmid incorporating the nucleic acid to be delivered and exosomes produced by the transfected cells are used; (2) a method of introducing nucleic acids into exosome from the outside by electroporation; and (3) a method of the nucleic acid modification of the lipid membranes of isolated exosomes via an anchor molecule. Exosomes are produced from various cell types, from plants to animals, and have a myriad of functions, including microRNAs and proteins with different compositions depending on the cell origin. It is expected that future developments will take advantage of the properties of various exosomes. A few biomaterials have been tested in clinical trials (Appendix A); exoASO-STAT6 is an exosome with STAT-6 ASO used for tumors in NCT05375604 [71]. iExosome is a mesenchymal stem cell (MSC)-derived exosome with KRAS G12D siRNA used for pancreatic cancer in NCT03608631 [72]. TTX-MC138 is dextran-coated iron oxide nanoparticle with BCL2L12 used for glioblastoma in NCT03020017 [73,74].

Biomaterials are useful for target delivery due to their biocompatibility and advanced functions based on bio-functional systems. However, biomaterials are difficult to synthesize chemically, but it is necessary to maintain the good quality and improve the stability, if needed, of biochemically synthesized materials.

## 5. Nucleic Acid Materials Based on the DNA Nanotechnology

DNA nanotechnology, established by Nadrian Seeman in the 1980s, forms nanostructures through the sequence-specific base pairing of DNA [75]. Nucleic acid delivery by incorporating nucleic acid medicine sequences into various nucleic acid nanostructures, including two-dimensionally arranged DNA origami, Y-shaped DNA, and three-dimensional pyramid-shaped and cube-shaped self-assemblies, is currently being studied as a nucleic acid delivery system (Figure 5a) [76,77].

DNA nanotechnology involving various nanostructure technologies, including chemical modification and material complex, has been reported [78]. Improving the efficiency of introducing nucleic acid structures into the cytoplasm remains challenging. Therefore, many studies have examined its use as an adjuvant via nucleic acid receptors in endocytosis. The CpG-ODN HEPLISAV-B^®^ [79] is used clinically as an adjuvant to activate the Nf-κB pathway via recognition by TLR9 in the endosomes of immune cells. The DNA nanostructures, DNA-tetrahedron nanostructure [80], DNA origami-tube nanostructure [81], polypod-like DNA nanostructure [82], and 3D-wireframe DNA origami nanostructures [83], which attempt to efficiently induce adjuvant activity, have been investigated. These nanostructures induce adjuvant activity higher than that of single-stranded CpG-ODNs. The use of chemically modified nucleic acids has improved the fragility of natural nucleic acids, and pyramid structures with mirror DNA (L-DNA) have demonstrated improved serum tolerance and significant accumulation in tumors [84]. Controlling the small size of the mirror-pyramid nanostructure has been shown to improve kidney cancer accumulation and achieve kidney cancer treatment [85].

As mentioned previously in Figure 4b, polysiRNAs are also polymerized via a terminal thiol group; it is a chemical modification-based nucleic acid structure [86]. Reversibly ionic-based nanoparticles (RIONs) of self-assembly nucleic acid are formed by sense strands of chemically modified cationic back bones and negatively charged ASs via base–base and electrostatic–static interactions (Figure 5b) [87]. RION caging antitumor microRNA-143 improved the blood retention in vivo, reduced the accumulation to the liver, and improved the accumulation in the tumor, thereby inducing antitumor effects (Figure 5b) [87]. Hybrid materials of nucleic acid with organic and inorganic substances, which nucleic acid located the surface, of nanoparticles have also progressed. Self-assembling nucleic acid nanoparticles in an aqueous solution are formed by nucleic acids attached to hydrophobic alkyl chains or cholesterol. Spherical nucleic acid (SNA), in which the coated nucleic acids of gold nanoparticle surfaces can be found, improve the in vivo stability of the nucleic acid due to steric hindrance with enzymes (Figure 5c) [88,89,90]. NU-0129 of SNA had completed the first human phase I clinical trials for recurrent glioblastoma and established the systemic treatment in NCT03020017 (Appendix A) [90]. Inorganic nanoparticles are monodispersed, and particles of various sizes can be designed. In addition to the size and surface charge of nanoparticles, their shape is also important for achieving good tissue dynamics and the good cellular uptake of nanoparticles [91,92,93]. Inorganic nanoparticles can be made into spheres or rods using template chemistry; it is expected that organ accumulation can be controlled using their shape. These nucleic acid nanostructures (NASs) have improved nuclease resistance, which are advantages compared with single- or double-strand nucleic acids. Improving the stability of NASs with negative charge repulsion within the structure in physiological conditions has been investigated. NASs are coated with several nanomaterials, lipids [94], proteins [95], and cationic polymers [96], increasing their stability. 

**Figure 5 pharmaceutics-16-01367-f005:**
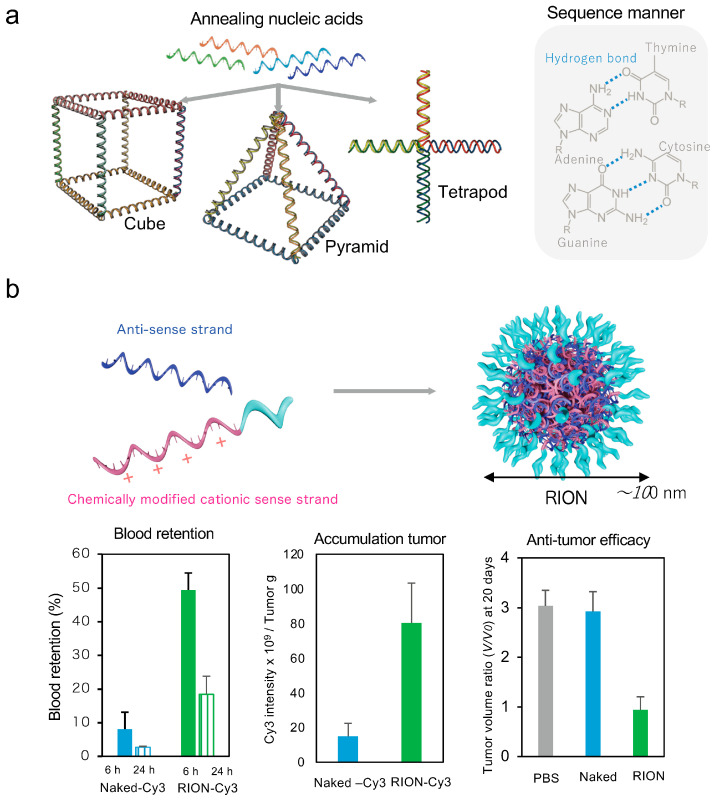
(**a**) Schematic of the DNA nanotechnology, (**b**) schematic of the reversibly ionic-based nanoparticles (RIONs) and their efficacy from ref. [87] (Permission obtained from Willy), and (**c**) schematic diagram of spherical nucleic acid (SNA) [90] (Permission obtained from The American Association for the Advancement of Science).

## 6. Clinical Trial in Japan

For the clinical trial status of nucleic acid medicines using nanotechnology in Japan (Table 3), a phase I trial of SRN-14, PRDM14 siRNA, and nanotechnology (GL2-800, branched PEG-polyornithine block polymer) in jRCT2031190181 was conducted for breast cancer patients. In jRCT2041230136, a phase I trial using TUG1ASO, noncoding RNA TUG1 ASO, and nanotechnology uPIC was performed for patients with recurrent glioblastoma. In jRCT2031200057, a phase I trial of the intertumoral administration of TDM-812, RPN2 siRNA, and nanotechnology AK6 was also carried out for treatment-resistant breast cancer patients [55]. In jRCT2061210058, a phase I trial of the intrathoracic administration of MIRX002, miR-3140-3p mimic, a mimic of gene suppressing microRNA, and nanotechnology AK6 was conducted for malignant pleural mesothelioma patients. Finally, in ACTRN12618001428257, the first human trial was performed by the systemic administration of NJA-730, CD40 siRNA, and nanotechnology using SPG. In this trial, no serious side effects caused by NJA-730 were reported [67].

The development progress in nanotechnology-based nucleic acid medicines has been demonstrated in past and current clinical trials outside of Japan (see Appendix A and Table 3), and 15 clinical trials, including 14 lipid-based nanotechnologies and 1 biomaterial-based nanotechnology, are ongoing. Three phase I trials are ongoing in Japan, and future results are awaited.

## 7. Conclusions

This review focused on the use of nanotechnology to accelerate the development of nucleic acid medicines and their clinical trials in Japan. Future research is expected to further diversify and maximize the usefulness of nanotechnology-based nucleic acid medicine. In addition to the development of conventional ASO, CpG-ODN, siRNA, miRNA, pDNA, and mRNA nucleic acid drugs, further cellular and biological regulation by DNA and RNA has been reported as follows: staple nucleic acids [97] for suppressing gene expression by the structural regulation of mRNA and the RNA sponge effect caused by the circular RNA (circRNA) [98,99] of noncoding RNA (ncRNA) are expected to further develop nucleic acid medicines in the areas of biological control and pharmaceutical applications. Additionally, research on nanotechnology with conjugate GalNAc, antibodies, and PEGylated compounds has been progressing. Although challenges in targeting the liver as well as ensuring quality control in the manufacturing of self-assembly nanotechnology remain, ongoing basic and clinical research studies will lead to breakthroughs and the development of nucleic acid medicines that target additional unmet medical needs.

## Figures and Tables

**Figure 1 pharmaceutics-16-01367-f001:**
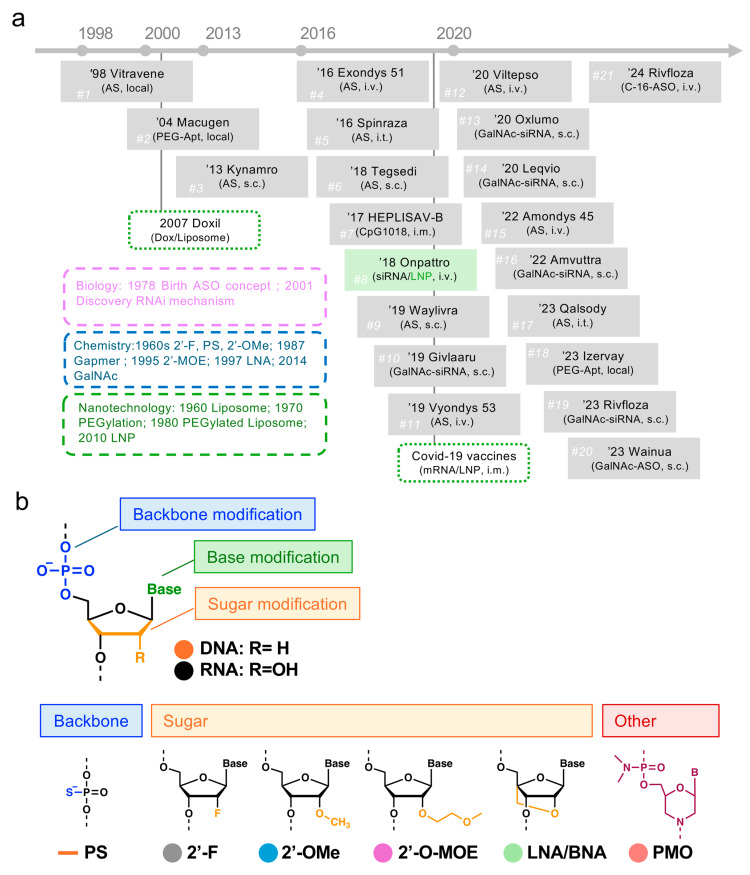
(**a**) List of approval nucleic acid medicines; (**b**) categorized chemical modification of nucleotide on backbone, sugar, or base; (**c**) design of chemically modified nucleic acid sequence. ASO, antisense oligonucleotide; siRNA, small interfering RNA; miRNA, microRNA; mRNA, messenger RNA; CpG-ODN, unmethylated cytosine guanine motif oligo deoxy nucleotide; i.v., intravenous injection; s.c., subcutaneous injection; i.m., intramuscular injection; i.t., intrathecal administration; PS, phosphorothioate; 2′-F, 2′-fluoro RNA; 2′-OMe, 2′-O-methylation RNA; 2′-MOE, 2′-O-methoxyethyl RNA; LNA/BNA, locked nucleic acid or bridged nucleic acid; GalNAc, N-acetylgalactosamine; PMO, morpholino oligomer; RISC, RNA-induced silencing complex; AS, antisense strand; SS, sense strand; STC, standard template chemistry; ESC, enhanced stabilization chemistry.

**Figure 2 pharmaceutics-16-01367-f002:**
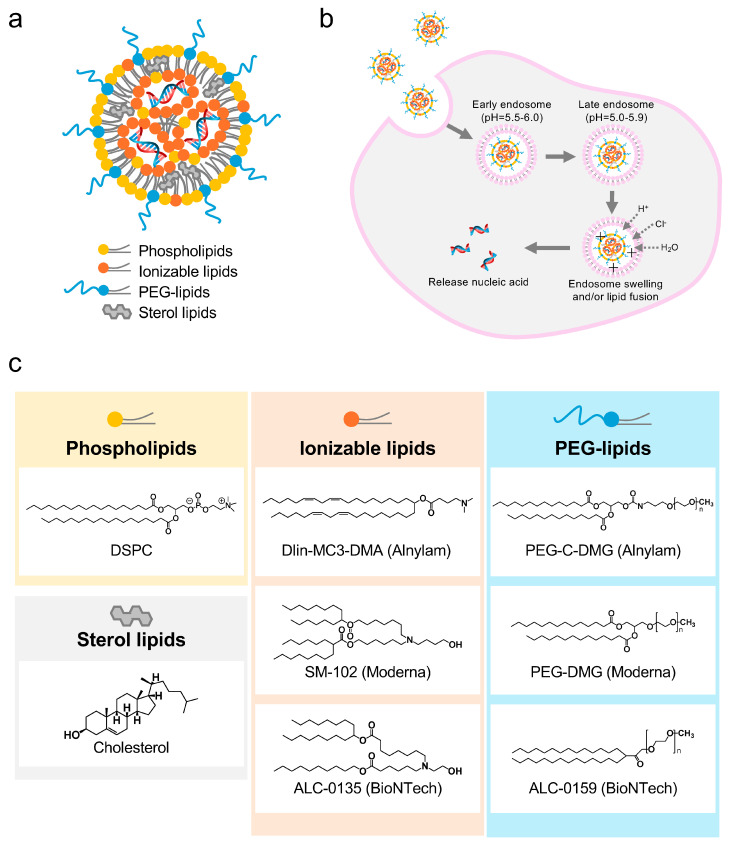
(**a**) Schematic of the formulation of lipid nanoparticles (LNPs), (**b**) a model of the endocytosis escape of the LNPs, (**c**) lipid composition of the approved LNPs, and (**d**) chemically modified mRNA.

**Figure 4 pharmaceutics-16-01367-f004:**
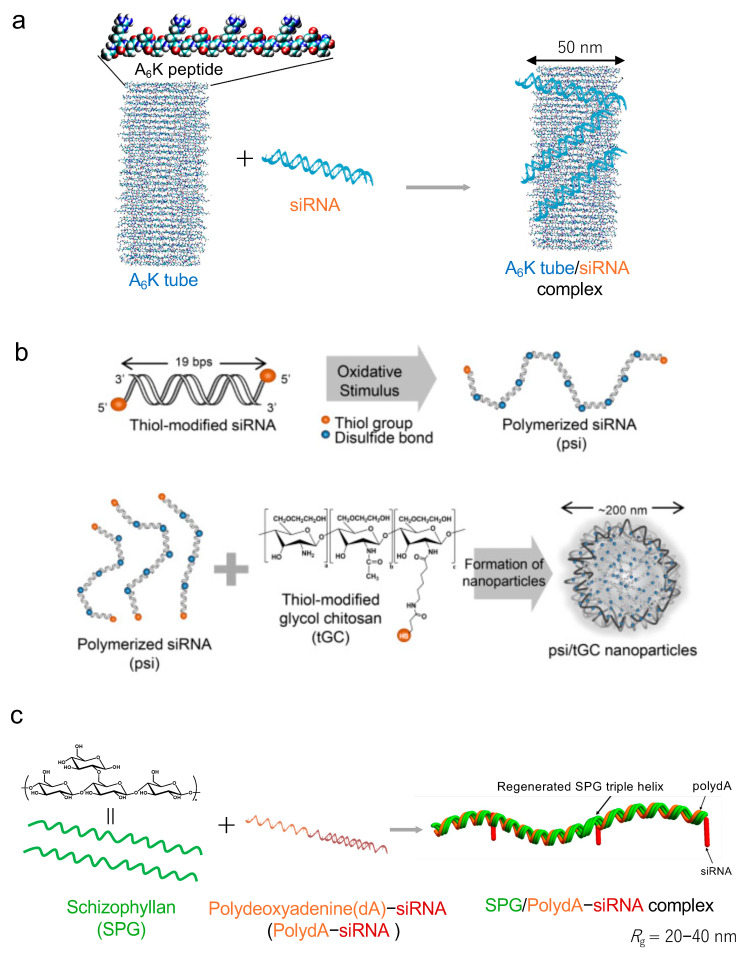
(**a**) Schematic of the A_6_K tube/siRNA complex by edited from ref. [54] (Permission obtained from Elsevier). (**b**) Schematic of the psi/tGC nanoparticles ref. [61] (Permission obtained from Elsevier). (**c**) Schematic of the SPG/PolydA-siRNA complex by edited from ref. [67] (Permission obtained from Elsevier). (**d**) Schematic of the exosome secretion (left) and formulation exosome (light). ncRNA, noncoding RNA; miRNA, microRNA; lncRNA, long ncRNA; circRNA, circular RNA.

**Table 1 pharmaceutics-16-01367-t001:** Lipid compositions of approved LNPs.

Drug Name	Molar Lipid Ratio (%) of the Helper Lipid/Cholesterol/PEGylated Lipid/Ionizable Lipid
Onpattro^®^	DSPC/Chol/PEG-lipid/Dlin-MC3-DMA = 10/38.5/1.5/50
Comirnaty^®^	DSPC/Chol/PEG-lipid/ALC-0315 = 9.4/42.7/1.6/46.3
Spikevax^®^	DSPC/Chol/PEG-lipid/SM-102 = 10/38.5/1.5/50

**Table 2 pharmaceutics-16-01367-t002:** List of organ-specific integrated lipid nanoparticles.

Organ-Specific Targeting	Lipid Compositions (mol/mol%)	Injection Route	Ref.
Liver	5A2-SC8/DOPE/Chol/PEG-lipid/DODAP = 19/19/38/3.8/20	i.v.	[34]
	5A2-SC8/DOPE/Chol/PEG-lipid = 35/16/46.5/2.5	i.v.	[36]
	Dlin-MC3-DMA/DSPC/Chol/PEG-lipid = 50/10/38.5/1.5	i.v.	
	C12-200/DOPE/Chol/PEG-lipid = 35/16/46.5/2.5	i.v.	
	306O_i10_/lipidoid/DOPE/Chol/PEG-lipid = 35/16/46.5/2.5	i.v.	[39]
	200O_i10_/lipidoid/DOPE/Chol/PEG-lipid = 35/16/46.5/2.5	i.v.	
	514O_6,10_/lipidoid/DOPE/Chol/PEG-lipid = 35/16/46.5/2.5	i.v.	
Spleen	5A2-SC8/DOPE/Chol/PEG-lipid/18PA(16.7/16.7/33.3/3.3/30	i.v.	[34]
	Dlin-MC3-DMA/DSPC/Chol/PEG-lipid/18PA = 35/7/27/1.1/30	i.v.	
	C12-200/DOPE/Chol/PEG-lipid/18PA = 24.5/11.2/32.6/1.8/30	i.v.	
	DOPE/Chol/C14-PEG-lipid/OF-Deg-Lin = 16/2.5/46.5/35	i.v.	[37]
	DNCA/CLD = 64.3/35.7	i.v.	[38]
Lung	5A2-SC8/DOPE/Chol/PEG-lipid/DOTAP = 11.9/11.9/23.8/2.4/50	i.v.	[34]
	5A2-SC8/DOPE/Chol/PEG-lipid/18:0EPC = 11.9/11.9/23.8/2.4/50	i.v.	[36]
	5A2-SC8/DOPE/Chol/PEG-lipid/18:1IR = 11.9/11.9/23.8/2.4/50	i.v.	
	Dlin-MC3-DMA/DSPC/Chol/PEG-lipid/DOTAP = 25/5/19.3/0.8/50	i.v.	[39]
	C12-200/DOPE/Chol/PEG-lipid/DOTAP = 17.5/8/23.3/1.3/50	i.v.	
Pancreas	306O_i10_/DOTAP/Chol/PEG-lipid = 35/40/22.5/2.5	i.p.	
	200O_i10_/DOTAP/Chol/PEG-lipid = 35/40/22.5/2.5	i.p.	
	514O_6,10_/DOTAP/Chol/PEG-lipid = 35/40/22.5/2.5	i.p.	

i.v., Intravenous; i.p., Intraperitoneal.

**Table 3 pharmaceutics-16-01367-t003:** Clinical trials of nanotechnology-based nucleic acid medicine in Japan.

Clinical Trial ID	Code No.	Nucleic Acid	Nanotechnology	Phase	Status	Ref.
jRCT2031190181	SRN-14	siRNA (PRDM14)	GL2-800 (Branched PEG-polyornithine block polymer)	I	Recruiting	
jRCT2041230136	TUG1ASO	ASO (ncRNA TUG1)	Cationic polymer, uPIC	I	Recruiting	
jRCT2031200057	TDM-812	siRNA (RPN2)	A6K	I	Suspended	[55]
jRCT2061210058	MIRX002	miRNA mimic (miR-3140-3p)	A6K	I	Recruiting	
ACTRN12618001428257	NJA-730	siRNA (CD40)	SPG	I	Completed	[67]

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
