# Peer review of "Advanced Nanotechnology-Based Nucleic Acid Medicines"

_pharmaceutics, 2024, doi:10.3390/pharmaceutics16111367_

Round 1

Reviewer 1 Report

Comments and Suggestions for Authors

This paper addresses these issues and advance of nucleic acid medicines, there is ongoing research about various nanotechnologies, including lipid nanoparticles (LNPs), synthetic polymer materials, biomaterials, and nucleic acid-based materials, which provides an overview of research on nucleic acid drugs utilizing nanotechnology. This paper is suggested to be published in Pharmaceutics after major revisions as following:

1. In the sentence: "LNP utilize lipids containing pH-responsive tertiary amines discovered by Peter Cullis", "pH-responsive" is very easy to be misunderstood as "pH-sensitive". Appropriate revision is suggested.

2. Thare are some reported works about nucleotidyl(cytidinyl)/cationic lipids encapsulated siRNA, ASO, aptamer and mRNA etc, which are suggested to be reviewed and enclosed in this paper.

Author Response

Comments 1: In the sentence: "LNP utilize lipids containing pH-responsive tertiary amines discovered by Peter Cullis", "pH-responsive" is very easy to be misunderstood as "pH-sensitive". Appropriate revision is suggested.

Response 1:Thank you for pointing this out. We re-consider and agree with this comment. Therefore, we have revised the“pH-responsve” to mention “an ionizable lipid”.

Comments 2: There are some reported works about nucleotidyl(cytidinyl)/cationic lipids encapsulated siRNA, ASO, aptamer and mRNA etc, which are suggested to be reviewed and enclosed in this paper.

Response 2: Thank you for your comments. We reviewed the nucleotidyl(cytidinyl)/cationic lipids as a specific delivery lipid in revised text and Table 2 and and the revised text as follows;

The lipoplex of neutral cytidinyl lipid, 2-(4-amino-2-oxopyrimidin-1-yl)-N-(2,3-dioleoyl-oxypropyl) acetamide (DNCA) and the cationic lipid, dioleoyl-3,3′-disulfanediylbis-[2-(2,6-diaminohexanamido)] propanoate (CLD) accumulates in the spleen via intravenous (i.v.) administration (Table 2)34.

Table 2. List of organ-specific integrated lipid nanoparticles

Organ-specific targeting

Lipid compositions (mol/mol %)

Injection route

Ref.

Liver

5A2-SC8/DOPE/Chol/DMG-PEG/DODAP=19/19/38/3.8/20

i.v.

[30]

5A2-SC8/DOPE/Chol/DMG-PEG=35/16/46.5/2.5

i.v.

[32]

Dlin-MC3-DMA/DSPC/Chol/DMG-PEG = 50/10/38.5/1.5

i.v.

C12-200/DOPE/Chol/DMG-PEG=35/16/46.5/2.5

i.v.

306Oi10/lipidoid/DOPE/Chol/PEG-lipid=35/16/46.5/2.5

i.v.

[35]

200Oi10/lipidoid/DOPE/Chol/PEG-lipid=35/16/46.5/2.5

i.v.

514O6/lipidoid/DOPE/Chol/PEG-lipid=35/16/46.5/2.5

i.v.

Spleen

5A2-SC8/DOPE/Chol/DMG-PEG/18PA(16.7/16.7/33.3/3.3/30

i.v.

[30]

Dlin-MC3-DMA/DSPC/Chol/DMG-PEG/18PA = 35/7/27/1.1/30

i.v.

C12-200/DOPE/Chol/DMG-PEG/18PA=24.5/11.2/32.6/1.8/30

i.v.

DOPE/Chol/C14-PEG-2000/OF-Deg-Lin=16/2.5/46.5/35

i.v.

[33]

DNCA/CLD=64.3/35.7

i.v.

[34]

Lung

5A2-SC8/DOPE/Chol/DMG-PEG/DOTAP= 11.9/11.9/23.8/2.4/50

i.v.

[30]

5A2-SC8/DOPE/Chol/DMG-PEG/18:0EPC=11.9/11.9/23.8/2.4/50

i.v.

[32]

5A2-SC8/DOPE/Chol/DMG-PEG/18:1IR=11.9/11.9/23.8/2.4/50

i.v.

Dlin-MC3-DMA/DSPC/Chol/DMG-PEG/DOTAP = 25/5/19.3/0.8/50

i.v.

[35]

C12-200/DOPE/Chol/DMG-PEG/DOTAP=17.5/8/23.3/1.3/50

i.v.

Pancreas

306Oi10/DOTAP/Chol/PEG-lipid=35/40/22.5/2.5

i.p.

200Oi10/DOTAP/Chol/PEG-lipid=35/40/22..5/2.5

i.p.

514O6/DOTAP/Chol/PEG-lipid=35/40/22.5/2.5

i.p.

References

34          L. Li, J. Long, Y. Sang, X. Wang, X. Zhou, Y. Pan, Y. Cao, H. Huang, Z. Yang, J. Yang, S. Wang, J. Control. Release 2021, 340, 114.

Reviewer 2 Report

Comments and Suggestions for Authors

In the current review, the advanced nanotechnology-based nucleic acid medicines, including typically LNP, synthetic polymer materials, biomaterials, and DNA-hybrid nucleic acid delivery nanotechnology, are widely discussed. Furthermore, clinical trials of nucleic acid medicines in Japan are summarized. The topic is interesting, but there are still several comments need to be addressed.

Question 1: Categorized chemical modification of nucleotide on backbone, sugar, or base are discussed in the Introduction. However, the discussions about these modification method seems simple. I am very interested about the modification methods, and the advantages and disadvantages between different location modification should be added in this section.

Question 2: As the most advanced nucleic acid delivery system, LNP has attracted generous research enthusiasm and achieved clinical translation currently. Typically ionizable lipids as you list, such as Dlin-MC3-DMA, SM-102, and ALC-0315, have extensively investigated. The clinical development about ionizable lipids not limited to the above-mentioned ionizable lipids should be listed as a table to help readers obtain more information.

Question 3: Selective Organ Targeting (SORT) technology plays an significant role in  improving LNP non-liver targeting distribution via optimizing the LNP composition with reduced workload. The design concept should be list in this paragraph. For example, while most ester-based ionizable lipids are reported to mainly transfect the liver, OF-Deg-Lin exhibits predominant splenic transfection. Further disclosure of the relationships between structure and organ selectivity would greatly benefit extrahepatic RNA delivery.

Question 4: In the section of “Synthetic polymer materials”, the characteristics of each material and the comparison between them should be added.

Question 5: Although the application for nucleic acid delivery of different materials has been discussed, there is a lack of summary of the advantages or disadvantages between different materials. The application scope or characteristics of different materials still lack discussions.

Author Response

Comments 1: Categorized chemical modification of nucleotide on backbone, sugar, or base are discussed in the Introduction. However, the discussions about these modification method seems simple. I am very interested about the modification methods, and the advantages and disadvantages between different location modification should be added in this section.

Response 1: Thank you for pointing this out. We agree with this comment. Therefore, we have revised text and add Figure. 1c as follows;

Approved nucleic acid drugs can be used for chemical modification of positions in nucleic acid sequences (Figure 1c). For the RNA-induced silencing complex (RISC), which requires an antisense strand (AS) and an Argonaute protein to recognize and cleave target mRNA, siRNA should be designed, and methods such as Standard Template Chemistry (STC) and Enhanced Stabilization Chemistry (ESC) modification methods have been proposed4–6.

Figure1. (c) The design of chemically modified nucleic acid sequence

4            A. Khvorova, J. K. Watts, Nat. Biotechnol. 2017, 35, 238.

5            T. C. Roberts, R. Langer, M. J. A. Wood, Nat. Rev. Drug Discov. 2020, 19, 673.

6            J. Maraganore, Nat. Biotechnol. 2022, 40, 641.

Comments 2: As the most advanced nucleic acid delivery system, LNP has attracted generous research enthusiasm and achieved clinical translation currently. Typically ionizable lipids as you list, such as Dlin-MC3-DMA, SM-102, and ALC-0315, have extensively investigated. The clinical development about ionizable lipids not limited to the above-mentioned ionizable lipids should be listed as a table to help readers obtain more information.

Response 2: Thank you for your comment. We agree with this comment. Therefore, we have revised text and add Figure. 1c as follows;

Nine-LNP medicines including optimized ionizable lipids are going in clinical trials (Supporting Table2)21,22.

Supporting Table. 2 List of ongoing clinical trials of LNP medicine (Oct-2024)

References

21          T. T. H. Thi, E. J. A. Suys, J. S. Lee, D. H. Nguyen, K. D. Park, N. P. Truong, Vaccines (Basel) 2021, 9, DOI 10.3390/vaccines9040359.

22          X. Hou, T. Zaks, R. Langer, Y. Dong, Nat Rev Mater 2021, 6, 1078.

Comments 3: Selective Organ Targeting (SORT) technology plays an significant role in  improving LNP non-liver targeting distribution via optimizing the LNP composition with reduced workload. The design concept should be list in this paragraph. For example, while most ester-based ionizable lipids are reported to mainly transfect the liver, OF-Deg-Lin exhibits predominant splenic transfection. Further disclosure of the relationships between structure and organ selectivity would greatly benefit extrahepatic RNA delivery.

Response 3: Thank you for your constructive comments. We reviewed the SORT technology and the design concept in revised text and Table 2 and the revised text as follows;

Selective Organ Targeting (SORT) LNPs were found to improve non-liver distribution via optimizing the lipids composition and control the surface charge of LNPs (Table 2) 30–32. LNPs with a neutral surface formed by the ionizable lipid 1,2-dioleoyl-3-dimethylammonium propane (DODAP) accumulate in the liver, those with a negatively charged surface formed by the anionic phospholipid 1,2-dioleoyl-sn-glycero-3-phosphate (18PA) accumulate in the spleen, and those with a positively charged surface formed by the cationic lipid 1,2-dioleoyl-3-trimethylammonium propane (DOTAP) accumulate in the lungs31. Lung-targeting SORT LNPs, which add a permanently cationic quaternary ammonium lipid to the fifth component (the lipid composition includes 40%–60% total lipids), are known to form different protein coronas of plasma proteins from liver-targeting LNPs (with reduced enrichment of ApoE) and immunoglobulins as well, with increased enrichment of vitronectin among other proteins. OF-Deg-Lin is an ionizable cationic lipid that accumulates in the spleen 33.

Different routes of nanoparticle administration affect protein corona formation and alter organ accumulation35. After i.v. or intraperitoneal (i.p.) administration LNPs containing cationic lipids accumulate in the liver and pancreas, respectively (Table 2)35.

Table 2. List of organ-specific integrated lipid nanoparticles

Organ-specific targeting

Lipid compositions (mol/mol %)

Injection route

Ref.

Liver

5A2-SC8/DOPE/Chol/DMG-PEG/DODAP=19/19/38/3.8/20

i.v.

[30]

5A2-SC8/DOPE/Chol/DMG-PEG=35/16/46.5/2.5

i.v.

[32]

Dlin-MC3-DMA/DSPC/Chol/DMG-PEG = 50/10/38.5/1.5

i.v.

C12-200/DOPE/Chol/DMG-PEG=35/16/46.5/2.5

i.v.

306Oi10/lipidoid/DOPE/Chol/PEG-lipid=35/16/46.5/2.5

i.v.

[35]

200Oi10/lipidoid/DOPE/Chol/PEG-lipid=35/16/46.5/2.5

i.v.

514O6/lipidoid/DOPE/Chol/PEG-lipid=35/16/46.5/2.5

i.v.

Spleen

5A2-SC8/DOPE/Chol/DMG-PEG/18PA(16.7/16.7/33.3/3.3/30

i.v.

[30]

Dlin-MC3-DMA/DSPC/Chol/DMG-PEG/18PA = 35/7/27/1.1/30

i.v.

C12-200/DOPE/Chol/DMG-PEG/18PA=24.5/11.2/32.6/1.8/30

i.v.

DOPE/Chol/C14-PEG-2000/OF-Deg-Lin=16/2.5/46.5/35

i.v.

[33]

DNCA/CLD=64.3/35.7

i.v.

[34]

Lung

5A2-SC8/DOPE/Chol/DMG-PEG/DOTAP= 11.9/11.9/23.8/2.4/50

i.v.

[30]

5A2-SC8/DOPE/Chol/DMG-PEG/18:0EPC=11.9/11.9/23.8/2.4/50

i.v.

[32]

5A2-SC8/DOPE/Chol/DMG-PEG/18:1IR=11.9/11.9/23.8/2.4/50

i.v.

Dlin-MC3-DMA/DSPC/Chol/DMG-PEG/DOTAP = 25/5/19.3/0.8/50

i.v.

[35]

C12-200/DOPE/Chol/DMG-PEG/DOTAP=17.5/8/23.3/1.3/50

i.v.

Pancreas

306Oi10/DOTAP/Chol/PEG-lipid=35/40/22.5/2.5

i.p.

200Oi10/DOTAP/Chol/PEG-lipid=35/40/22..5/2.5

i.p.

514O6/DOTAP/Chol/PEG-lipid=35/40/22.5/2.5

i.p.

References

30          Qiang Cheng, Tuo Wei, Lukas Farbiak, Lindsay T. Johnson, Sean A. Dilliard, Daniel J. Siegwart, Nat. Nanotechnol. 2020, 15, 313.

31          S. A. Dilliard, Q. Cheng, D. J. Siegwart, Proc. Natl. Acad. Sci. U. S. A. 2021, 118, e2109256118.

32          S. A. Dilliard, Y. Sun, M. O. Brown, Y.-C. Sung, S. Chatterjee, L. Farbiak, A. Vaidya, X. Lian, X. Wang, A. Lemoff, D. J. Siegwart, J. Control. Release 2023, 361, 361.

33          O. S. Fenton, K. J. Kauffman, J. C. Kaczmarek, R. L. McClellan, S. Jhunjhunwala, M. W. Tibbitt, M. D. Zeng, E. A. Appel, J. R. Dorkin, F. F. Mir, J. H. Yang, M. A. Oberli, M. W. Heartlein, F. DeRosa, R. Langer, D. G. Anderson, Adv. Mater. 2017, 29, 1606944.

35          J. R. Melamed, S. S. Yerneni, M. L. Arral, S. T. LoPresti, N. Chaudhary, A. Sehrawat, H. Muramatsu, M.-G. Alameh, N. Pardi, D. Weissman, G. K. Gittes, K. A. Whitehead, Sci. Adv. 2023, 9, eade1444.

Comments 4:  In the section of “Synthetic polymer materials”, the characteristics of each material and the comparison between them should be added.

Response 4: Thank you for your comment. We agree with this comment. Therefore, we have revised text and Figure 2 as follows;

The nanoparticle has a diameter of 75 nm and binds to the transferrin receptor of the target. CALAA-01 had undergone its first clinical trials for solid tumor in NCT00689065 (Supporting table 3).

Unit PIC (uPIC) is formed by a Y-shaped polymer, a branched block copolymer comprising PEG, polylysine, and siRNA. The diameter of uPIC is 18 nm.

Figure 3. (a) Schematic diagram of the CALAA-01 by edited from ref [38] (Permission obtained from American Chemical Society) (b) Schematic diagram of the uPIC by edited from ref [41](Permission obtained from Elsevier)

Comments 5: Although the application for nucleic acid delivery of different materials has been discussed, there is a lack of summary of the advantages or disadvantages between different materials. The application scope or characteristics of different materials still lack discussions.

Response 5:  Thank you for valuable your comments. We revised text as follows;

Section 2

Although research on LNPs has progressed to clinical findings and has shown no-liver accumulation in vivo, the nucleic acid loading ratio in a nanoparticle is low at <10%, whereas the “PEG dilemma” antibodies cause faster phagocytosis, which is called the accelerated blood clearance (ABC) phenomenon, and is a future challenge 36,37.

Synthetic polymers are expected to help reduce the cost of pharmaceuticals because they can be synthesized in large quantities, but the process must comply with the Chemistry, Manufacturing, and Control (CMC) guidance that carefully regulates the distribution of polymers.

Section 4

Biomaterials are useful for target delivery due to their biocompatibility and advanced functions based on bio-functional systems. However, biomaterials are difficult to synthesize chemically, but it is necessary to maintain good quality and improve stability, if needed, of biochemically synthesized materials.

Reviewer 3 Report

Comments and Suggestions for Authors

The manuscript entitled “Advanced nanotechnology-based nucleic acid medicines” aims to review the latest advances in nanotechnology to obtain nucleic acids medicines. The manuscript includes updated bibliography in this field, which is to be positively valued, but there are some aspects that authors should clarify before accepting for publication:

1.      In the introduction section author mention 19 approved nucleic acids medicines, but it is not clear whether they talk about any medicine containing a nucleic acid as active moiety or they only refer to ASOs, because in the following lines they give examples of modifications applied only to ASOs. Moreover they use several abbreviations that are not defined.

RNases enzymes are mentioned as possible cause of degradation, but it should be more appropriate to talk about nucleases; depending on the nature of the nucleic acid nuclease responsible for degradation can be RNases or DNases.

Authors should take into account that nucleic acids can be considered mRNA or plasmid DNA for supplementation gene therapy, siRNA, ASOs, microRNA, for silencing… Most of them are mentioned in the manuscript, but the introduction does not include them. Along the text authors mention some types of nucleic acids (for example ncRNAs miR, lncRNA or circRNA in figure 4 and conclusions) that has not been previously mentioned or described.

Regarding DNA, authors mention this nucleic acid as adjuvant in section 6, but not as active molecule properly in the manuscript. Plasmid DNA has been extensively formulated in different types of nanosystems (lipids, polymers, peptides, …).

I encourage the authors to include a section describing the types of nucleic acids they consider in the review and why those and no others are the objective of present work.

Moreover, why does section 6 include examples about inorganic nanoparticles that do not implicate the use of DNA?

2.      In general, the content of the manuscript is rather scarce. Sections 3 and 4 are too brief. Authors should justify why they have only included those few works in those sections. A section explaining the methodology used for the bibliographic search would be appreciated (keywords, databases, inclusion and exclusion criteria, …)

3.      Authors should also justify why they have only included in the work clinical trials registered in Japan. Since Pharmaceutics is an international journal consulted by researchers from different regions worldwide, inclusion of a more complete table with examples from other regions would be enriching for researchers and for the journal.

4.      Figure and table legends should describe all the abbreviations included in each figure and table. Likewise, I recommend authors to revise the text to describe all abbreviations used throughout the manuscript, the first time they appear in the text.

5.      References in bibliography section lack of titles; please update.

Author Response

Comments 1: In the introduction section author mention 19 approved nucleic acids medicines, but it is not clear whether they talk about any medicine containing a nucleic acid as active moiety or they only refer to ASOs, because in the following lines they give examples of modifications applied only to ASOs. Moreover they use several abbreviations that are not defined.

RNases enzymes are mentioned as possible cause of degradation, but it should be more appropriate to talk about nucleases; depending on the nature of the nucleic acid nuclease responsible for degradation can be RNases or DNases.

Authors should take into account that nucleic acids can be considered mRNA or plasmid DNA for supplementation gene therapy, siRNA, ASOs, microRNA, for silencing… Most of them are mentioned in the manuscript, but the introduction does not include them. Along the text authors mention some types of nucleic acids (for example ncRNAs miR, lncRNA or circRNA in figure 4 and conclusions) that has not been previously mentioned or described.

Regarding DNA, authors mention this nucleic acid as adjuvant in section 6, but not as active molecule properly in the manuscript. Plasmid DNA has been extensively formulated in different types of nanosystems (lipids, polymers, peptides, …).

I encourage the authors to include a section describing the types of nucleic acids they consider in the review and why those and no others are the objective of present work.

Moreover, why does section 6 include examples about inorganic nanoparticles that do not implicate the use of DNA?

Response 1:Thank you for valuable your comments. We revised the text as follows;

Section 1

…Nucleic acid medicines include antisense oligo nucleic acid (ASOs), small interfering RNA (siRNA), and microRNA (miRNA), which are gene silencing, mRNA, plasmid DNA, which are supplementation gene therapy, an unmethylated cytosine guanine motif oligonucleotide(CpG-ODN), which are immune activation for adjuvant.

Nucleic acids are rapidly degraded by nuclease enzymes including DNase and RNase in the blood…

Section 6

…Hybrid materials of nucleic acid with organic and inorganic substances, which nucleic acid located the surface, of nanoparticle, has also progressed.…

Comments 2:   In general, the content of the manuscript is rather scarce. Sections 3 and 4 are too brief. Authors should justify why they have only included those few works in those sections. A section explaining the methodology used for the bibliographic search would be appreciated (keywords, databases, inclusion and exclusion criteria, …)

Other clinical trials are summarized in supporting Table 2. Local Drug EluteR (LODERTM) is the a biodegradable polymer poly(lactic-co-glycolic) acid (PLGA) with KRAS(G12D) siRNA for pancreatic cancer treatment in NCT167625941. MK-4621 is an cationic polymer polyethyleneimine (PEI, JetPEITM) with CpG-ODN for tumor in NCT0373913842. STP705 is histidine-lysine co polymer (HKP) with two types of siRNAs of PTGS2 and TGF- b for tumor in NCT05196373. STP707 is HKP with two types of siRNAs of Cox-2 and TGF-b for tumor in NCT0503714943.

  1. Indini, A., Rijavec, E., Ghidini, M., Cortellini, A. & Grossi, F. Targeting KRAS in solid tumors: Current challenges and future opportunities of novel KRAS inhibitors. Pharmaceutics 13, (2021).
  2. Moreno, V. et al. Treatment with a retinoic acid-inducible gene I (RIG-I) agonist as monotherapy and in combination with pembrolizumab in patients with advanced solid tumors: results from two phase 1 studies. Cancer Immunol. Immunother. 71, 2985–2998 (2022).
  3. Kim, W. et al. Codelivery of TGFβ and Cox2 siRNA inhibits HCC by promoting T-cell penetration into the tumor and improves response to Immune Checkpoint Inhibitors. NAR Cancer 6, zcad059 (2024).

A few  biomaterials have been clinical trial in supporting Table 2, exoASO-STAT6 is an exosome with STAT-6 ASO for tumor in NCT0537560467. iExosomes is a mesenchymal stem cell (MSC)-derived exosomes with KRAS G12D siRNA for pancreatic cancer in NCT0360863168. TTX-MC138 is dextran-coated ion oxide nanoparticle with BCL2L12 for glioblastoma in NCT0302001769,70.

  1. Kamerkar, S. et al. Exosome-mediated genetic reprogramming of tumor-associated macrophages by exoASO-STAT6 leads to potent monotherapy antitumor activity. Sci. Adv. 8, eabj7002 (2022).
  2. Mendt, M. et al. Generation and testing of clinical-grade exosomes for pancreatic cancer. JCI Insight 3, (2018).
  3. Medarova, Z., Robertson, N., Ghosh, S., Duggan, S. & Varkaris, A. Initial clinical experience with the first-in-class anti-metastasis therapeutic TTX-MC138. J. Clin. Oncol. 42, e15072–e15072 (2024).
  4. Medarova, Z. et al. Abstract PO5-27-06: Development of TTX-MC138, a first-in-Class miRNA-10b-Targeted Therapeutic Against Metastatic Cancers of Diverse Primary Disease Origins. Cancer Res. (2024) doi:10.1158/1538-7445.sabcs23-po5-27-06.

Comments 3: Authors should also justify why they have only included in the work clinical trials registered in Japan. Since Pharmaceutics is an international journal consulted by researchers from different regions worldwide, inclusion of a more complete table with examples from other regions would be enriching for researchers and for the journal

Response 3: Thank you for your constructive comments. We reviewed abstract and added supporting table 2 as more worldwide understanding as follow;

Abstract

…The past and current clinical trial development over 30 products in international (including Europe and USA) and two clinical trials are ongoing. In Japan, three Phase I trials are ongoing, and future results are awaited. This review in Advanced Pharmaceutical Science and Technology in Japan as a topical collection summarizes the latest research in the nanotechnology of nucleic acid medicines and status of clinical trial in Japan, which is expected to further evolve in the future.

Supporting table. 3 List of clinical trials of nanotechnology-based nucleic acid medicine outside Japan in past to present (Oct-2024)

Comments 4:  Figure and table legends should describe all the abbreviations included in each figure and table. Likewise, I recommend authors to revise the text to describe all abbreviations used throughout the manuscript, the first time they appear in the text.

Response 4: Thank you for your comments. We revised all the bloviations in the text and added supporting table 1 for abbreviations.

Supporting table. 1 List of abbreviations

Comments 5:  References in bibliography section lack of titles; please update.

Response 5: Thank you for your comments. We update the references with titles.

Reviewer 4 Report

Comments and Suggestions for Authors

The manuscript is focused on a very pertinent and actual topic, the development of nucleic acid medicines through nanotechnology, highlighting several relevant aspects. The manuscript is well written and discussed in some extent and deserves to be published. However, I feel some points can be further addressed and clarified, and, therefore, the manuscript can be improved.

Following are the critiques to revise the manuscript prior to its publication.    

Q1. The key-words only include mention to RNA. Key-words containing “DNA” should also be considered to broad the review subject.  

Q2. Also, in the Key-words, may be included “lipid nanoparticles” as this can increase the interest of the audience and highly capture the attention of readers, researchers…

Q3. The authors should better clarify the main subject and message of the presented review, as well as give future perspectives on what and how should researchers pay attention and evolve to progress in nanotechnology-based nucleic acid medicines.  

Q4. The abstract should mention, at least, in one sentence what is the position of Japan concerning the advances in such nanotechnologies in comparison to the rest of the World. For instance, a comparison to Europe and USA, can be made to give the reader a more global/full picture, and to broad the audience.

Q5. Please revise Table 1 related to text content. It seems not well co-related with the references provided. Moreover, references should be added to the Table.  

Q6. Page 3, line 80. What is “nucleic acid-biased bled”? Is this a typing error? If so, please revise and correct. Another example, page 9, line 242: What do the authors mean with “therapeutic nucleic medicine”?

Q7. Except for Figure 4, overall, the illustrations of the figures are cheap. It is a waste because the content is rich. I think more eye-catching illustrations should be used.

Q8. The authors need to carefully revise the entire manuscript for possible grammatical errors corrections.

Q9. The authors should revise the paper carefully for English language style, grammar and spelling, and make appropriate corrections and changes. 

Comments on the Quality of English Language

The authors should revise the paper carefully for English language style, grammar and spelling, and make appropriate corrections and changes. 

Author Response

Comments 1: The key-words only include mention to RNA. Key-words containing “DNA” should also be considered to broad the review subject.

Comments 2: Also, in the Key-words, may be included “lipid nanoparticles” as this can increase the interest of the audience and highly capture the attention of readers, researchers…

Response 1 and 2: Thank you for your comment. We revised the keywords as follow;

Nucleic acid medicine, siRNA, miRNA, mRNA, DNA, ASO, CpG-ODN, pDNA, Drug delivery system, Nanotechnology, Nanotechnology-based medicine, Lipid nanoparticles, LNP

Comments 3: The authors should better clarify the main subject and message of the presented review, as well as give future perspectives on what and how should researchers pay attention and evolve to progress in nanotechnology-based nucleic acid medicines.  

Response 3: Thank you for valuable your comments. We revised text as follows;

Conclusions

This review focused on nanotechnology for accelerating the development of nucleic acid medicines and their clinical trials in Japan. Future research is expected to further diversify and maximize the usefulness of nanotechnology-based nucleic acid medicine. In addition to the development of conventional ASO, CpG-ODN, siRNA, miRNA, pDNA and mRNA nucleic acid drugs, further cellular and biological regulation by DNA and RNA has been reported: staple nucleic acids93 for suppressing gene expression by structural regulation of the mRNA and the RNA sponge effect caused by the circular RNA (circRNA)94,95 of noncoding RNA(ncRNA) are expected to further develop nucleic acid medicine in the areas of biological control and pharmaceutical applications. Additionally, research on nanotechnology with conjugate GalNAc, antibodies, and PEGylated compounds has been progressing. Although challenges in targeting the liver as well as ensuring quality control in the manufacturing of self-assembly nanotechnology, ongoing basic and clinical research studies will lead to breakthroughs and the development of nucleic acid medicines that target additional unmet medical needs.

Comments 4: The abstract should mention, at least, in one sentence what is the position of Japan concerning the advances in such nanotechnologies in comparison to the rest of the World. For instance, a comparison to Europe and USA, can be made to give the reader a more global/full picture, and to broad the audience.

Response 4: Thank you for valuable your comments. We revised abstract as follows;

Abstract

The past and current clinical trial development over 30 products in international (including Europe and USA) and two clinical trials are ongoing. In Japan, three Phase I trials are ongoing, and future results are awaited.

Comments 5: Please revise Table 1 related to text content. It seems not well co-related with the references provided. Moreover, references should be added to the Table 3.  

Table 3. Clinical trial of nanotechnology-based nucleic acid medicine in Japan

Comments 6: Page 3, line 80. What is “nucleic acid-biased bled”? Is this a typing error? If so, please revise and correct. Another example, page 9, line 242: What do the authors mean with “therapeutic nucleic medicine”?

Response 6: Thank you for your comments. We revised these words in text as follow;

…the nucleic acid-based bleb…

nucleic acid medicine…

Comments 7:  Except for Figure 4, overall, the illustrations of the figures are cheap. It is a waste because the content is rich. I think more eye-catching illustrations should be used.

Response 7: Thank you for pointing this out. We re-consider and revised Figures including Figure 4 in revised text as follow;

Figure1. (a) List of approval nucleic acid medicines, (b) Categorized chemical modification of nucleotide on backbone, sugar, or base. (c) The design of chemically modified nucleic acid sequence.

ASO, Antisense oligonucleotide; siRNA, small interfering RNA; miRNA or miRNA, microRNA; mRNA, messenger RNA; CpG-ODN, unmethylated cytosine guanine motif oligo deoxy nucleotide; PS, phosphorothioate; 2'-F, 2'-Fluoro RNA; 2'-OMe,          2'-O-methylation RNA; 2'-MOE, 2'-O-methoxyethyl RNA; LNA/BNA, Locked nucleic acid or Bridged nucleic acid; GalNAc, N-acetylgalactosamine; PMO, Morpholino oligomer; RISC, RNA-induced silencing complex; AS, Antisense strand; SS, Sense strand; STC, Standard Template Chemistry; ESC, Enhanced Stabilization Chemistry

Figure2. (a) Schematic diagrams of the formulation of lipid nanoparticles (LNPs) (b) A model of endocytosis escape of the LNPs (c) Lipids composition of the approved-LNPs.

Figure 3. (a) Schematic diagram of the CALAA-01 by edited from ref 38 (Permission obtained from American Chemical Society) CDP, Cationic cyclodextrin-based polymer; AD-PEG, Adamantane-PEG; AD-PEG-Tf, AD-PEG-Transferrin (b) Schematic diagram of the uPIC by edited from ref 44(Permission obtained from Elsevier).

Figure 4. (a) Schematic diagram of the A6K tube/siRNA complex by edited from ref [29] (Permission obtained from Elsevier) (b) Schematic diagram of the psi/tGC nanoparticles ref [34](Permission obtained from Elsevier). (c)Schematic diagram of the SPG/PolydA-siRNA complex by edited from ref [37] (Permission obtained from Elsevier). (d)Schematic diagram of the exosome secretion (left) and formulation exosome (light). ncRNA, noncoding RNA; miRNA, microRNA; lncRNA, long-ncRNA; circRNA, circular RNA

Figure 5. (a) Schematic diagram of the DNA nanotechnology (b) Schematic diagram of the Reversibly ionic-based nanoparticles (RIONs) and its efficacy from ref 83(Permission obtained from Willy). (c) Schematic diagram of the Spherical nucleic acid (SNA) from ref 86(Permission obtained from The American association for the advancement of science).

Comments 8:  The authors need to carefully revise the entire manuscript for possible grammatical errors corrections.

Comments 9: The authors should revise the paper carefully for English language style, grammar and spelling, and make appropriate corrections and changes. 

Response 8 and 9: Thank you for valuable your comments. The revised text was edited with Native English Proofreading service (Enago corporation) carefully.

Round 2

Reviewer 2 Report

Comments and Suggestions for Authors

They should add and disscuss some other references of gene delivey systems.

Author Response

Thank you again for taking the time to review this manuscript. Please find the response below and the corresponding revisions in the re-submitted files.

Comment:
They should add and disscuss some other references of gene delivey systems.

Response:
Thank you for valuable your comments. We revised text as follows.

Section 2

…Fourteen-LNP medicines, including optimized ionizable lipids with mRNAs, gene-editing RNAs, pDNAs, and ASOs, are currently in the clinical trial stage (Supporting Table 2)23,24. The gene-editing RNAs use CRISPR/Cas9 technology to guide RNA and Cas9 mRNA25,26.

Reviewer 3 Report

Comments and Suggestions for Authors

I thank the authors for taking my comments into consideration. The article has improved considerably, but before accepting it I think they should address the following issues:

1.       In the new text, English should be revised. See some examples bellow, but the work requires a deeper revision.

a.       For example, please revise this sentence in abstract; is something missing?: “The past and current clinical trial development over 30 products in international (including Europe and USA) and two clinical trials are ongoing.”

b.       In the following sentence in the introduction section “Approved nucleic acid drugs can be used for chemical modification of positions in nucleic acid sequences”, should “for” be replace by “by”?

c.       The sentence “ASOs are designed in the form gapmer type, which is 10 bases of DNA are placed in the center and nucleic acid analogs are placed at both ends with 2'-MOE modification (Figure 1c).” I also difficult to understand. Please revise.

d.       In the sentence “MK-4621 is an cationic polymer polyethyleneimine (PEI, JetPEITM)…” “an” should be replaced by “a”.

e.       Rewrite the following sentence: “A few biomaterials have been clinical trial in supporting Table 3”

f.        TTX-MC138 is dextran-coated ion oxide nanoparticle: replace ion by iron

2.       In the introduction section authors now mention different types of nucleic acids, but they only explain modifications in siRNA and ASO. However, modifications in nucleotides of mRNA are usually key factors to optimize its efficacy and immunogenicity. Although the review is focused on nanotechnology as an indispensable tool for the advances in nucleic acids-based medicines, it would be appreciated to include a section after introduction talking about the importance of the modifications of all the mentioned nucleic acids, and explaining them deeper.

3.       In table 2: define i.v. and i.p.

4.       In the title3 of Table authors should replace “Clinical trial” by “Clinical trials”.

Author Response

Thank you again for taking the time to review this manuscript. Please find the detailed responses below and the corresponding revisions in the re-submitted files.

Comments: I thank the authors for taking my comments into consideration. The article has improved considerably, but before accepting it I think they should address the following issues:

Comment. 1      In the new text, English should be revised. See some examples bellow, but the work requires a deeper revision.
a.       For example, please revise this sentence in abstract; is something missing?: “The past and current clinical trial development over 30 products in international (including Europe and USA) and two clinical trials are ongoing.”
b.       In the following sentence in the introduction section “Approved nucleic acid drugs can be used for chemical modification of positions in nucleic acid sequences”, should “for” be replace by “by”?
c.       The sentence “ASOs are designed in the form gapmer type, which is 10 bases of DNA are placed in the center and nucleic acid analogs are placed at both ends with 2'-MOE modification (Figure 1c).” I also difficult to understand. Please revise.
d.       In the sentence “MK-4621 is an cationic polymer polyethyleneimine (PEI, JetPEITM)…” “an” should be replaced by “a”.
e.       Rewrite the following sentence: “A few biomaterials have been clinical trial in supporting Table 3”
f.        TTX-MC138 is dextran-coated ion oxide nanoparticle: replace ion by iron

Response1: Thank you for your valuable comments. The text was revised according to your comments and re-edited carefully with a Native English Proofreading service (Enago corporation).

a.…In countries other than Japan (including Europe and the USA), >40 nanotechnology-based nucleic acid medicines have been tested in clinical trials, and 15 clinical trials are ongoing.…

b.…Approved nucleic acid medicines can be used by chemical modification of positions in nucleic acid sequences (Figure 1c).…

c.

…ASOs are designed as gapmer and exon-skipping types. The structure of the gapmer type consists of 2'-MOE introduced at both ends of the sequence and natural DNA in the center (Figure 1c).…

d.…MK-4621 is a cationic polymer polyethyleneimine (PEI, JetPEITM)…

  1. …A few biomaterials have been tested in clinical trials (Supporting Table 3)…

f. …TTX-MC138 is dextran-coated iron oxide nanoparticle with BCL2L12 for glioblastoma in NCT0302001773,74.…

Comments. 2       In the introduction section authors now mention different types of nucleic acids, but they only explain modifications in siRNA and ASO. However, modifications in nucleotides of mRNA are usually key factors to optimize its efficacy and immunogenicity. Although the review is focused on nanotechnology as an indispensable tool for the advances in nucleic acids-based medicines, it would be appreciated to include a section after introduction talking about the importance of the modifications of all the mentioned nucleic acids, and explaining them deeper.

Response. 2 Thank you for your valuable comments. We revised the text as follows.

Section 1

…For the RNA-induced silencing complex (RISC), which requires an antisense strand (AS) of siRNA and an argonaute protein to recognize and cleave target mRNA, siRNA should be designed with 2'-F and 2'-OMe, and methods such as standard template chemistry (STC) and enhanced stabilization chemistry (ESC) modification methods have been proposed1,4–6. 2'-F and 2'-OMe is composed of 2' modification of sugar moiety. The introduction of 2'-F and 2'-OMe improve nuclease resistance and target RNA binding.

ASOs are designed as gapmer and exon-skipping types. The structure of the gapmer type consists of 2'-MOE introduced at both ends of the sequence and natural DNA in the center (Figure 1c). The 2'-MOE portion is introduced for nuclease resistance, and the natural DNA portion is involved in recognition and cleavage of the target RNA. The other ASO in Figure 1c, the exon-skipping type, is chemically modified with PMO. PMO comprises a six-membered morpholine sugar moiety and a phosphorodiamidate phosphate moiety; moreover, it is electrically neutral. PMO binds to the target RNA but is not cleaved by RNase H, thereby regulating gene expression without cleavage.…

Comments. 3 and 4:
3. In table 2: define i.v. and i.p.
4. In the title3 of Table authors should replace “Clinical trial” by “Clinical trials”.

Response. 3 and 4 Thank you for your comments. We revised the text as follows.

Response. 3

…Table 2. List of organ-specific integrated lipid nanoparticles

i.v., Intravenous ;i.p., Intraperitoneal…

Response. 4

Table 3. Clinical trials of nanotechnology-based nucleic acid medicine in Japan

Reviewer 4 Report

Comments and Suggestions for Authors

The authors have followed all my comments and suggestions, and answered to all the raised points. With this approach, they have further clarified the main issues of the manuscript and improve the quality of figures, Tables and captions. Also, the authors made changes to the text concerning English languague. 

I do recommend the publication of the manuscript in this current form.

Comments on the Quality of English Language

The authors made an effort to improve the quality of English language. I do recommend minor English editing.

Author Response

Thank you very much for taking the time to review this manuscript. Please find the detailed responses below and the corresponding revisions in the re-submitted files.

Comments: The authors have followed all my comments and suggestions, and answered to all the raised points. With this approach, they have further clarified the main issues of the manuscript and improve the quality of figures, Tables and captions. Also, the authors made changes to the text concerning English languague. 

I do recommend the publication of the manuscript in this current form. Comments on the Quality of English Language The authors made an effort to improve the quality of English language. I do recommend minor English editing.

Response: Thank you for your valuable comments. The revised text was carefully re-edited by a Native English Proofreading service (Enago corporation).